# CP-BCS: Binary Code Summarization Guided by Control Flow Graph and Pseudo Code

Tong Ye[1], Lingfei Wu[2], Tengfei Ma[3], Xuhong Zhang[1], Yangkai Du[1]
Peiyu Liu[1], Shouling Ji[1], Wenhai Wang[1*],

[1]Zhejiang University; [2]Anytime.AI; [3]Stony Brook University
{tongye,zhangxuhong,yangkaidu,liupeiyu,sji,zdzzlab}@zju.edu.cn
lwu@anytime-ai.com, tengfei.ma@stonybrook.edu

## Abstract

Automatically generating function summaries for binaries is an extremely valuable but challenging task, since it involves translating the execution behavior and semantics of the low-level language (assembly code) into human-readable natural language. However, most current works on understanding assembly code are oriented towards generating function names, which involve numerous abbreviations that make them still confusing. To bridge this gap, we focus on generating complete summaries for binary functions, especially for stripped binary (no symbol table and debug information in reality). To fully exploit the semantics of assembly code, we present a control flow graph and pseudo code guided binary code summarization framework called CP-BCS. CP-BCS utilizes a bidirectional instruction-level control flow graph and pseudo code that incorporates expert knowledge to learn the comprehensive binary function execution behavior and logic semantics. We evaluate CP-BCS on 3 different binary optimization levels (O1, O2, and O3) for 3 different computer architectures (X86, X64, and ARM). The evaluation results demonstrate CP-BCS is superior and significantly improves the efficiency of reverse engineering.

## 1 Introduction

Most commercial off-the-shelf software is closed-source and typically distributed as stripped binaries that lack a symbol table or any debug information (e.g., variable names, function names). This practice is mainly done for easy distribution, copyright protection, and malicious evasion. Professionals seeking to analyze these stripped binaries must perform reverse engineering and inspect the logic at the binary level. While current binary disassemblers, such as IDA Pro (Hex-Rays, 2021) and Ret-Dec (Avast Software, 2021), can translate machine code into assembly code, the assembly representation only consists of plain instruction mnemonics

with limited high-level information, making it difficult to read and understand, as shown in Figure 1. Even an experienced reverse engineer needs to spend a significant amount of time determining the functionality of an assembly code snippet.

```
1.    mov rax,  [rdi]
2.    test rax, rax
3.    jz short loc_FFFFFFFF81752D04
4.    push rbp
5.    mov rbp, rsp
6.    push rbx
7.    mov rbx, rdi
8.    mov rdi, [rax+8]
9.    test rdi, rdi
10.   jz short loc_FFFFFFFF81752CE2
11.   mov rax, [rax]
12.   mov rax, [rax+48h]
13.   call qword ptr [rax+28h]
14.   mov rax, [rbx]
15.   mov rdi, [rax]
16.   call sub_FFFFFFFF8175287F
17.   mov rdi, [rbx]
18.   call sub_FFFFFFFF8114766E
19.   mov qword ptr [rbx], 0
20.   mov eax, 0
21.   pop rbx
22.   pop rbp
23.   retn
```

Figure 1: A sample of assembly code. The function name is: **gss_del_sec_context**. The summary is: **free all resources associated with context_handle**.

To mitigate this issue, researchers have made initial attempts, with recent studies focusing on predicting function names of binaries (Gao et al., 2021; Jin et al., 2022; Patrick-Evans et al., 2023). Function name prediction is the process of automatically generating a function name for a given assembly code snippet, which aims at showing the high-level meaning of the function. As shown in Fig 1, the target function name is *"gss_del_sec_context"*. Although some progress has been achieved in function name prediction, function names themselves can only partially and superficially represent the semantics of assembly code. Furthermore, function names frequently contain various abbreviations and custom tokens defined by developers (e.g., *"gss"*, *"del"*, *"sec"* in the example above). Consequently,

---

*  Corresponding author.

relying solely on function names can make it difficult to obtain an accurate description of an assembly code snippet and may even cause confusion.

We argue that generating a high-quality descriptive summary is a more direct and fundamental approach to strike at the essence compared to function name prediction (e.g., *"free all resources associated with context_handle."* in the Figure 1). Similar tasks have been extensively studied at the source code level (known as source code summarization) for languages such as Python and Java (LeClair et al., 2020; Shi et al., 2021; Wu et al., 2021; Guo et al., 2022c). At the source code level, researchers typically rely on advanced code analysis tools to extract fine-grained code structure properties and leverage the high-level semantic information inherent in the source code itself. However, assembly code, a low-level language, often lacks high-level, human-readable information and is prone to ambiguity. Moreover, the absence of fine-grained assembly code analysis tools makes it more challenging to gain a semantic understanding of assembly code. Furthermore, we discover that current large language models, such as ChatGPT (OpenAI, 2022), generally possess only a rudimentary understanding of assembly code without high-level abstract semantic comprehension (shown in Appendix A).

In this paper, we specifically concentrate on binary code summarization in stripped scenarios, which is a highly practical setting, and present CP-BCS. The novel CP-BCS framework comprehensively represents the execution behavior and semantics of assembly code from three different perspectives inspired by observing how human engineers analyze assembly code in practice. (1) **Assembly Instruction**: the assembly instructions themselves provide certain features such as memory operations and setting of register values. In addition, we also take into account some meaningful strings that remained in stripped binaries, such as the name of externally called function. (2) **Control Flow Graph**: to obtain the logical execution order of the assembly code, we extract the Control Flow Graph (CFG) of the assembly code. Considering the order relationship between adjacency instruction, we augment the original basic-block level CFG to a Bidirectional Instruction-level Control Flow Graph (BI-CFG). (3) **Pseudo Code**: due to the difficulty of understanding assembly code, plugins that attempt to decompile assembly code into high-level, C-like language (known as Pseudo

Code) are available. Although many of the resulting pseudo codes are imprecise and usually cannot be compiled, it still encompasses expert knowledge and understanding derived from human reverse engineers. Furthermore, considering the lack of meaningful high-level strings in pseudo code in realistic scenarios, inspired by pre-trained models such as CodeT5's (Wang et al., 2021) great performance in source code-related tasks, we explore the potential of utilizing such pre-trained models to recover missing semantic strings in pseudo code on stripped binaries. Our objective is to further narrow the gap between pseudo code and natural language by leveraging the capability of pre-trained models.

To facilitate further research in this area, we have made our dataset and code publicly available [1]. In summary, the contributions of this paper can be outlined as follows:

- To the best of our knowledge, CP-BCS is the first system for practically stripped binary code summarization. CP-BCS fully learns the execution behavior and semantics preserved in binary functions from three perspectives.

- We manually construct a comprehensive dataset, which is the first dataset to include {*assembly code, summary*} pairs for three different computer architectures (X86, X64, and ARM) and three different optimization levels (O1, O2, and O3).

- We conduct extensive experiments to evaluate the effectiveness of CP-BCS. The results on both automatic metrics and human evaluation demonstrate the superiority of CP-BCS. In particular, the human evaluation indicates that CP-BCS can significantly improve the efficiency of reverse engineers' comprehension of binary functions.

## 2   Related Works

**Function Name Prediction in Binary.**   Function name prediction is a task for binaries aimed at generating binary function names. NFRE (Gao et al., 2021) proposes two data-preprocessing approaches to mitigate the ambiguity of function names. SYMLM (Jin et al., 2022) proposes a neural architecture by learning context-sensitive behavior-aware code embedding. However, they still have

---

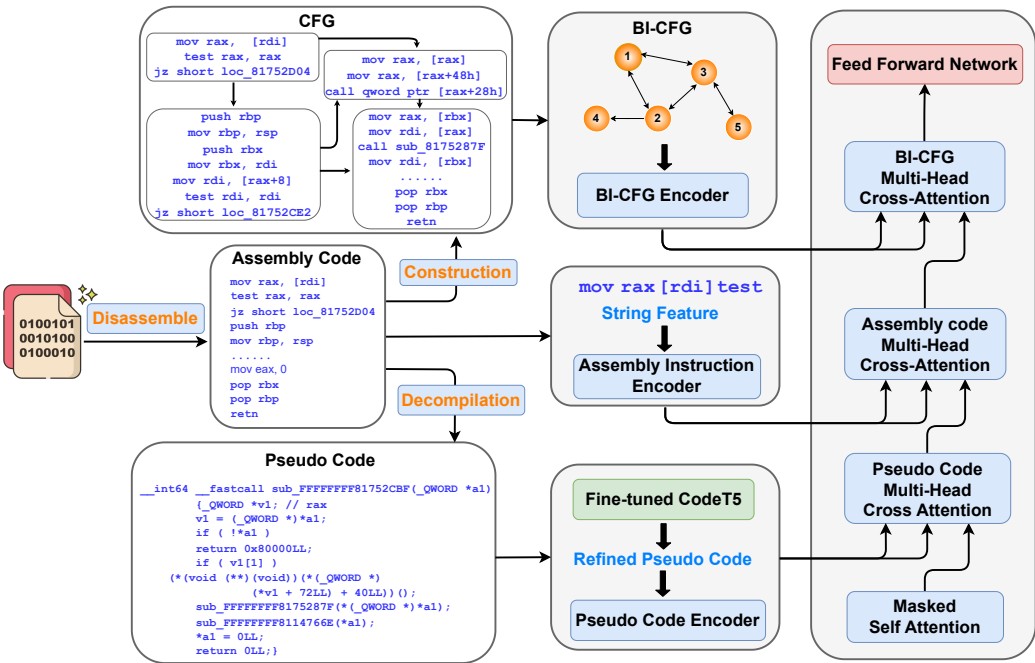

Figure 2: The overall architecture of CP-BCS.

not solved the ambiguous function name issues. XFL (Patrick-Evans et al., 2023) performs multi-label classification and learns an XML model to predict common tokens found in the names of functions from C binaries in Debian. As XFL predicts labels instead of whole function names, it is able to predict names for functions even when no function of that name is contained in the training set. The biggest difference between them and us is that we directly generate function summary sentences rather than a few function name tokens.

**Source Code Summarization.** Both binary and source code summarization aim to generate a concise and human-readable summary of a given code snippet. However, there are many sophisticated tools available for source code, such as parsers and token-level code analysis tools, which can help with the summarization process. Based on these tools, many approaches propose exploiting source code's structural properties, including Abstract Syntax Tree, Program Dependency Graph in a hybrid way (Iyer et al., 2020; Choi et al., 2021; Shi et al., 2021; Zhu et al., 2022), or structured-guided way (Son et al., 2022; Guo et al., 2022c; Ye et al., 2023). In contrast, binary code analysis tools are much coarser and can only achieve basic functionalities such as block jumping and function cross-referencing. In summary, source code summarization is generally easier due to the

human-readable nature of the code, preservation of information, and availability of tools. Binary code summarization, on the other hand, is more challenging due to its lower-level representation, loss of information, and ambiguity.

## 3 Methodology

### 3.1 Overview

Our proposed CP-BCS framework is designed as a plugin in the disassembler, such as IDA Pro or an online service[2], which automatically generates a human-readable descriptive summary for stripped functions. The whole architecture of CP-BCS is presented in Figure 2. As a prerequisite, the stripped binary is disassembled into assembly code by IDA Pro, and the functions in the binary are correctly recognized. The assembly code is then input into CP-BCS, which is essentially an encoder-decoder architecture, to ultimately generate the corresponding summary. CP-BCS consists of three encoders (*Assembly Instruction Encoder*, *BI-CFG Encoder*, and *Pseudo Code Encoder*) and a summary decoder. Next, we elaborate on the principle and implementation of CP-BCS.

### 3.2 Assembly Instruction Encoder

To understand the semantics of binary functions, the assembly code itself is the first-hand source that

---

[2]http://www.binarycodesummarization.com

can be utilized. It composes of a series of instructions, each of which is responsible for performing an action, such as reading and writing register or memory addresses. Each instruction is composed of an opcode (e.g., mov, add) and one or more operands (e.g., rax, [rdi]). We treat each opcode and operand as a separate token. This is because each opcode or operand carries its own semantic information, and we aim to learn the semantics of each word as finely as possible rather than treating the entire instruction as one token, like in binary function name prediction (Gao et al., 2021).

Although stripped binaries lack symbol tables and debugging information, we have found that there is still some string information in the assembly code, such as the names of externally called functions, which we called *string features*. These *string features* provide additional high-level information that can help to some extent in understanding the behavior of the assembly code.

We input the assembly tokens and *string features* into the Assembly Instruction Encoder (AIEnc). The AIEnc is essentially a Transformer encoder (Vaswani et al., 2017), which consists of stacked multi-head attention and parameterized linear transformation layers. Each layer emphasizes on self-attention mechanism. Considering that the semantic representation of the opcode and operand does not rely on the absolute positions, instead, their mutual interactions influence the meaning of the assembly code. To achieve this, we adopt a relative position encoding (Shaw et al., 2018) instead of an absolute position to better learn the semantic representation of each assembly token. The assembly code snippet is assumed to consist of $p$ tokens $[t_1, t_2, ..., t_p]$, after AIEnc, each token has a corresponding semantic representation, which is denoted as:

$$[h_1, h_2, ..., h_p] = AIEnc([t_1, t_2, ..., t_p])$$

### 3.3 BI-CFG Encoder

In order to better understand the structure and execution behavior of assembly code, we extract the Control Flow Graph. A canonical CFG is comprised of basic blocks and jump control flows. The nodes portray basic blocks, and the edges portray jump control flows, as shown in the upper left corner of Figure 2. However, it should be noted that canonical CFGs are based on basic blocks, which overlook the sequential execution relationships between adjacent instructions within basic blocks.

Further, traditional CFGs are unidirectional, which means each instruction cannot receive information from the instruction executed after it. To address these limitations, we propose a Bidirectional Instruction-level Control Flow Graph (BI-CFG). BI-CFG treats each instruction as a node and incorporates the logical execution order between instructions, as well as the jumps control flow between basic blocks, achieving a level of granularity at the instruction level. Furthermore, BI-CFG allows each instruction to aggregate node features from both forward and backward instructions, enabling bidirectional processing.

To improve the representation ability of BI-CFG, advanced graph neural networks are adopted to achieve this goal. Taking advantage of the GAT's (Veličković et al., 2018) exceptional performance and its ability to assign adaptive attention weights to different nodes, we employ GAT Encoder (GATEnc) to represent each node in the BI-CFG. The GATEnc layer processes the BI-CFG by first aggregating the neighbors of the instruction nodes with edge information. It then updates the instruction nodes with the aggregated information from their neighborhoods. After updating the node information, the node representations are put together into a $ReLU$ activation followed by residual connection (He et al., 2016) and layer normalization (Ba et al., 2016). Assuming the BI-CFG contains $q$ instruction nodes $[n_1, n_2, ..., n_q]$, after the GATEnc, each node has a semantic representation:

$$[r_1, r_2, ..., r_q] = GATEnc([n_1, n_2, ..., n_q])$$

### 3.4 Pseudo Code Encoder

Considering that assembly code is extremely low-level and hard to comprehend, there is a large gap between it and natural language summary. However, plugins are available that can facilitate the comprehension of assembly code by decompiling it into pseudo code. Compared to assembly code, pseudo code is a higher-level C-like language and can narrow the gap and alleviate the difficulty for reverse engineers to analyze assembly code. Although the generated pseudo code is not precise and often cannot be compiled, it still embodies expertise and comprehension derived from human reverse engineers. We believe that integrating pseudo code with expert knowledge can facilitate a more comprehensive comprehension of the semantics of assembly code from an alternative perspective.

However, in real-world stripped scenarios,

pseudo code often lacks meaningful strings, such as variable and function names, which are replaced by placeholders. This inspires us to explore ways to recover these missing strings as much as possible. With the emergence of pre-trained models, such as CodeT5 (Wang et al., 2021), Unixcoder (Guo et al., 2022a), which have demonstrated remarkable performance on source code-related tasks, we are motivated to consider utilizing the pre-trained models' comprehension of source code and natural language to recover the missing semantic strings in pseudo code to the fullest extent possible. To achieve this goal, we take the pseudo code decompiled from the stripped binary as input and the corresponding pseudo code decompiled from the non-stripped binary as the target to fine-tune CodeT5, as shown in Figure 3. We expect the fine-tuned CodeT5 can recover meaningful strings in the original pseudo code, such as function names, variable names, and other comments, etc. Following such recovery, the original pseudo code is enriched with more high-level string content, which we refer to as *refined pseudo code*.

For *refined pseudo code*, we employ an additional encoder, known as Pseudo Code Encoder (PSEnc), that is identical to the AIEnc for representation learning. Assuming the *refined pseudo code* contains $n$ tokens $[p_1, p_2, ..., p_n]$, after PSEnc, each token has a semantic representation, which is denoted as:

$$[v_1, v_2, ..., v_n] = PSEnc([p_1, p_2, ..., p_n])$$

### 3.5 Summary Decoder

The summary decoder is designed with modified Transformer decoding blocks. At time step $t$, given the existing summary tokens $[s_1, s_2, ..., s_{t-1}]$, the decoding blocks first encode them by masked multi-head attention. After that, we expand the Transformer block by leveraging three multi-head cross-attention modules to interact with the three encoders for summary decoding, as shown on the right side in Figure 2. A multi-head cross-attention module is applied to the pseudo code token features to obtain the first-stage decoded representation. This representation is then passed through another multi-head cross-attention module over the learned assembly token features for the second-stage decoding, which is further fed into the third multi-head cross-attention module over the learned instruction node features for the third-stage decoding. Then the decoded summary vectors are put

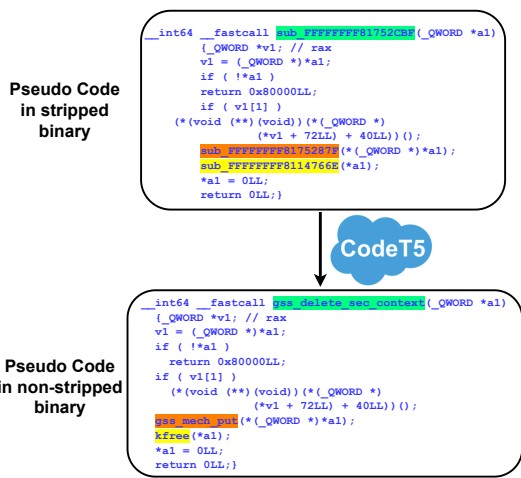

Figure 3: Fine-tune CodeT5 using Pseudo Code from stripped binary and corresponding Pseudo Code from non-stripped binary.

into a feed-forward network for non-linear transformation.

## 4 Dataset Construction and Statistics

### 4.1 Dataset Construction

It is non-trivial to obtain high-quality datasets for binary code summarization in the stripped scenario. The construction process of the entire dataset is shown in Figure 4.

① **Preliminary Survey.** We conduct a preliminary investigation with 15 reverse engineers from academia and industry to explore the types of binaries that reverse engineers encounter in their daily work, as well as other related questions (further details can be found in Appendix B). Additionally, we also include binaries commonly utilized in other binary-related tasks, such as binary clone detection (Ding et al., 2019; Yang et al., 2022). In total, we identify 51 corresponding binary projects in real-world scenarios. The specific binary projects are listed in Appendix C.

② **Source Code Collection.** Based on the preliminary survey, we collect these 51 binary projects and their corresponding source code from Github or their official websites.

③ **Compiled Source Code.** We manually compile these binary projects using the compiler (gcc-7.3.0) into three different optimization levels (O1, O2, O3) for three different computer architectures (X86, X64, ARM). It is noted that each binary file contains nine different variants.

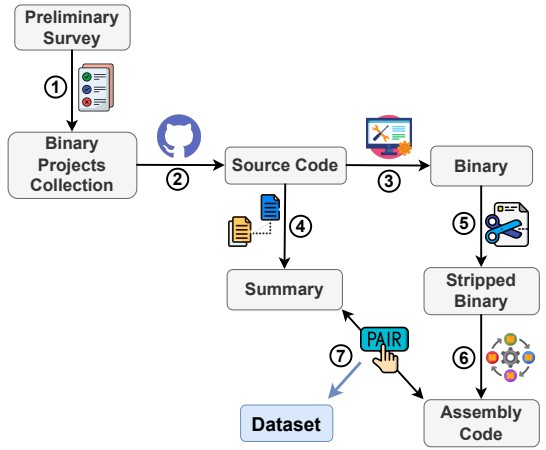

Figure 4: The construction process of the dataset.

⑦ **Making of Pairs.** Initially, we use the *function_boundaries* as indices to assign the function name to the function in the stripped binary. Next, we use *function_name* as indices to connect the summary and the corresponding stripped assembly code together. Finally, we construct pairs in the format of {*stripped assembly code, summary*}, which forms instances of the final **Dataset**.

## 4.2 Dataset Statistics

| Datasets (Arch: X64) | O1 | O2 | O3 |
|---|---|---|---|
| Train | 12,801 | 11,949 | 10,812 |
| Validation | 1,600 | 1,494 | 1,351 |
| Test | 1,599 | 1,493 | 1,351 |
| Assembly Code: Avg. tokens | 213.08 | 222.61 | 316.47 |
| BI-CFG: Avg. nodes | 42.57 | 44.83 | 57.54 |
| BI-CFG: Avg. edges | 62.78 | 69.14 | 93.84 |
| Pseudo Code: Avg. tokens | 228.65 | 243.32 | 359.99 |
| Summary: Avg. tokens | 9.74 | 9.58 | 9.68 |

Table 1: Dataset statistics for X64 architecture with (O1, O2, and O3) optimization levels.

Table 1 displays the statistics of three datasets under three optimization levels on the X64 architecture. Each specific architecture and optimization level corresponds to a specific dataset. The statistics of the datasets for two other architectures (X86, ARM) and some additional explanations about the datasets can be found in Appendix D.

## 5 Experiments

### 5.1 Experimental Setup

**Out-of-Vocabulary.** The vast operators in assembly code may produce a much larger vocabulary than natural language, which can cause Out-of-Vocabulary problem. To avoid this problem, inspired by related studies (Gao et al., 2021; Patrick-Evans et al., 2023), we empirically set the following rules to normalize assembly code:

- Retaining all the mnemonics and registers.

- Replacing all the constant values with <Positive>, <Negative> and <Zero>.

- Replacing all internal functions with <ICall>.

- Replacing all the destinations of local jump with <JumpAddress>.

**Metrics.** Similar to source code summarization, we evaluate the binary code summarization performance using three widely-used metrics, BLEU (Papineni et al., 2002), METEOR (Banerjee and Lavie,

④ **Summary Extraction.** We extract separate function-summary pairs from the source code. Specially, we extract functions and the associated comments marked by special characters "/**" and "*/" over the function declaration. These comments can be considered as explanations of the functions. We filter comments inside the function, and the first sentence was selected as the summary, which is consistent with the approach used in extracting summary in the source code summarization domain (Hu et al., 2018a; Liu et al., 2021). As a result, we get {*function_name, summary*} tuples.

⑤ **Binary Stripping.** To ensure consistency with the real stripped scenario, we employ the *"strip -s"* command to strip the binary. The strip operation removes sections such as "debug", "symtable", "strtab", etc., resulting in the elimination of symbol tables and all debugging information from the binary file.

⑥ **Binary Disassembling.** We use IDA Pro (Hex-Rays, 2021) to disassemble the original binary and the stripped binary to obtain their corresponding assembly code. We then separate the assembly code at the function level. For the assembly code from the original binary, we extract tuples in the form of {*function_name, function_boundaries*}. However, in the stripped binary, the function name is replaced by a placeholder *sub_address*, but the function boundaries remain unchanged whether or not the binary is stripped. For the assembly code from the stripped binary, we extract triplets in the form of {*sub_address, stripped assembly code, function_boundaries*}.

2005) and ROUGE-L (Lin, 2004). Furthermore, to provide a more accurate reflection of actual performance, we have designed a human evaluation that includes three aspects: **Similarity** (the similarity between CP-BCS generated summary and the ground-truth), **Fluency** (the fluency level of the results generated by CP-BCS) and **Time-Cost** (to what extent our model can improve the efficiency of reverse engineering). Further details on the human evaluation are deferred to Appendix E.

**Training Details.** We implement our approach based on NVIDIA 3090. The batch size is set to 32 and Adam optimizer is used with an initial learning rate $10^{-4}$. The training process will terminate after 100 epochs or stop early if the performance on validation set does not improve for 10 epochs. In addition, we leverage greedy search during validation and beam search (Koehn, 2004) during model inference and set beam width to 4.

## 5.2 Main Results

| ARCH | OPT | BLEU | ROUGL-L | METEOR |
|------|-----|------|---------|--------|
| ARM | O1 | 29.75 | 27.84 | 16.81 |
| ARM | O2 | 29.56 | 27.67 | 15.98 |
| ARM | O3 | 26.66 | 24.26 | 14.03 |
| Avg. | - | 28.66 | 26.59 | 15.61 |
| X86 | O1 | 26.57 | 25.04 | 13.50 |
| X86 | O2 | 25.74 | 23.74 | 13.34 |
| X86 | O3 | 26.38 | 25.04 | 13.24 |
| Avg. | - | 26.23 | 24.60 | 13.36 |
| X64 | O1 | 26.86 | 26.62 | 14.59 |
| X64 | O2 | 25.50 | 23.64 | 12.70 |
| X64 | O3 | 25.14 | 23.92 | 13.30 |
| Avg. | - | 25.83 | 24.73 | 13.53 |

Table 2: CP-BCS overall performance across different architectures (ARCH) and optimizations (OPT).

We first evaluate the overall performance of CP-BCS on our datasets. As shown in Table 2, the BLEU metric falls within the range of 20-30, indicating that "the gist is clear, but has grammatical errors" according to Google interpretation[3] of BLEU. Besides, there are two interesting findings: (1) **CP-BCS performs better on the ARM architecture compared to X86 and X64.** On average, CP-BCS on ARM outperforms X86 and X64 by

---

[3]https://cloud.google.com/translate/automl/docs/evaluate. Intervals of 10-19 indicate that the summary is "hard to get the gist", while intervals of 30-40 mean the summary is "understandable to good translations".

2.43 and 2.83 BLEU points, respectively. This is attributed to the simpler and more flexible Reduced Instruction Set Computing (RISC) architecture of ARM, while X86 and X64 rely on the Complex Instruction Set Computing (CISC) with a larger number of operation codes and registers to support complex mathematical operations, making it more challenging for CP-BCS to understand their assembly codes. (2) **CP-BCS performs better under the O1 optimization level compared to O2 and O3.** Through our empirical observation of assembly code under different optimization levels, the O2 and O3 optimization levels employ abundant advanced techniques such as vectorization instructions and loop unrolling to improve program execution speed but generate more complex assembly code. By contrast, O1 uses simpler methods, such as register allocation and basic block reordering, without generating overly complex assembly code, which can also be reflected in dataset statistics in Table 1. Thus, the assembly code generated by O1 is relatively simpler and easier for CP-BCS to extract semantic features.

## 5.3 Baselines and Ablation Study

| Model | BLEU | ROUGL-L | METEOR |
|-------|------|---------|--------|
| Assembly Code Only | 22.88 | 18.82 | 11.09 |
| Pseudo Code (CodeT5) | 22.89 | 22.04 | 11.89 |
| Pseudo Code (CodeT5+) | 24.14 | 23.83 | 12.48 |
| Pseudo Code (UniXcoder) | 23.17 | 22.65 | 12.35 |
| CP-BCS w/o Pseudo Code | 24.50 | 21.54 | 12.35 |
| CP-BCS w/o BI-CFG | 24.37 | 21.75 | 12.53 |
| CP-BCS w/o Refined | 25.61 | 23.20 | 13.12 |
| **CP-BCS (Full Model)** | **26.86** | **26.62** | **14.59** |

Table 3: Baselines and ablation study results on the dataset for X64 architecture with O1 optimization level.

**Baselines.** While binary function name prediction methods exist and have made processes, such as mitigating the ambiguity of function names (Gao et al., 2021) and converting to multi-label classification (Patrick-Evans et al., 2023), their entire workflow and goals differ greatly from our task. Therefore, it is difficult to directly compare the performance of these methods with our approach. We adopt "Assembly Code Only" and "Pseudo Code" as our baselines. The formal solely uses assembly code to generate the summary, while the latter uses the corresponding pseudo code and summary pairs to fine-tune pre-trained models, such as CodeT5

(Wang et al., 2021), CodeT5+ (Wang et al., 2023) and UniXcoder (Guo et al., 2022b). We select these two classes as baselines because they are the most straightforward and intuitive ways to tackle the task.

**Ablation Study.** To evaluate the effectiveness of CP-BCS components, we conduct a set of ablation studies. We design three models for comparison, each one removing an important component from CP-BCS, as follows: (1) remove BI-CFG, labeled as *CP-BCS w/o BI-CFG*; (2) remove pseudo code, labeled as *CP-BCS w/o Pseudo Code*; (3) keep pseudo code but without refined, labeled as *CP-BCS w/o Refined*. For demonstration purposes, we choose a dataset with a specific architecture (X64) and optimization level (O1). The ablation experiment results for other architectures and other optimization levels (the remaining eight groups) are included in Appendix F. As shown in Table 3, the performance of CP-BCS is affected when any of these components are removed. The result of *CP-BCS w/o BI-CFG* and *CP-BCS w/o Pseudo Code* show that the BI-CFG and pseudo code are the most significant learning components of CP-BCS. Removing BI-CFG and pseudo code resulted in a performance decrease of 2.49 and 2.36 BLEU points, respectively. Moreover, the performance of *CP-BCS w/o Refined* indicates that *refined pseudo code* can further enhance the performance of CP-BCS; a detailed case is shown in Section 5.6. Similar conclusions can be drawn from the ablation experiments on other datasets, further demonstrating the universality of the three important components.

## 5.4 Human Evaluation

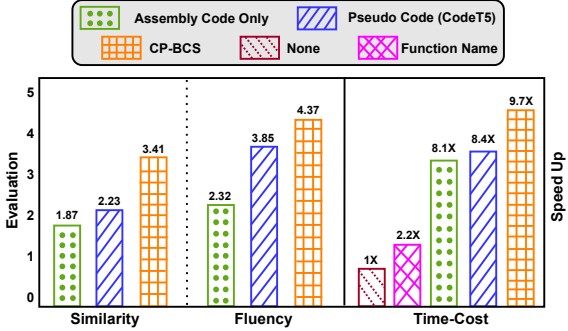

Figure 5: Human evaluation. "Assembly Code Only" and "Pseudo Code (CodeT5+)" are the two baselines. "None" means only given assembly code; "Function Name" means given assembly code and the corresponding function name.

We conduct a human evaluation (details provided in Appendix E) to assess the quality of the generated summaries by CP-BCS in terms of **Similarity**, **Fluency**, and **Time-Cost**, as depicted in Figure 5. The results on the similarity and fluency metrics show that CP-BCS can generate summaries that are more similar to the ground truth and more fluent in naturalness. Moreover, the time-cost results indicate that CP-BCS significantly enhances the efficiency of reverse engineers' comprehension of assembly code. In particular, compared to the "None" scenario (only given assembly code), CP-BCS improves speed by 9.7 times.

## 5.5 Study on the Model Structures

In this section, we evaluate the performance of CP-BCS across varied model structures. Specially, we investigate the impact of the sequencing among three distinct cross-attention modules in the summary decoder on the final performance. Furthermore, we explore the implications of directly concatenating assembly code with pseudo code and using a single encoder for representation.

| Cross-attention Module Orders | BLEU | ROUGL-L | METEOR |
|---|---|---|---|
| assembly code→BI-CFG→pseudo code | 26.71 | 26.37 | 14.45 |
| assembly code→pseudo code→BI-CFG | 26.50 | 26.40 | 14.39 |
| BI-CFG→assembly code→pseudo code | 26.45 | 25.95 | 14.31 |
| BI-CFG→pseudo code→assembly code | 26.47 | 26.08 | 14.49 |
| pseudo code→assembly code→BI-CFG | **26.86** | **26.62** | 14.59 |
| pseudo code→BI-CFG→assembly code | 26.86 | 26.45 | **14.67** |

Table 4: Different cross-attention module orders on the dataset for X64 architecture with O1 optimization level.

| ARCH:X64; OPT:O1 | BLEU | ROUGL-L | METEOR |
|---|---|---|---|
| concat (assembly + pseudo) | 24.49 | 21.71 | 12.68 |
| concat (assembly + pseudo) + BI-CFG | 25.83 | 24.33 | 13.55 |
| CP-BCS | **26.86** | **26.62** | **14.59** |

Table 5: Directly concatenation of assembly code and pseudo code on the dataset for X64 architecture with O1 optimization level.

Table 4 presents the performance of different orders (first→second→third) among the three distinct cross-attention modules (assembly code, BI-CFG, and pseudo code) in the summary decoder on the dataset for X64 architecture with O1 optimization level. The results shows that different orders only have a slight impact on the final performance (the BLEU score did not fluctuate by more than 0.5 points). In Table 5, we use "concat (assembly + pseudo)" to present directly concatenating

assembly code with pseudo code. The results show that using a single encoder to represent the concatenated body of assembly code and pseudo code can degrade the model's final performance. Therefore, assigning a separate encoder for assembly code and pseudo code is a better choice.

### 5.6 Case Study of Refined Pseudo Code

| | |
|---|---|
| *Pseudo Code (Stripped)* | ```int __fastcall sub_E6D18(_DWORD *a1) { if (a1[55] != dword_162354)     return 0; sub_E6C4C(a1); --dword_162354; return 1; }``` |
| *Refined Pseudo Code* | ```int __fastcall burn_drive_free_subs (burn_drive *d) { if (d->sub.nodep.cnt !=     subs_allocated)     return 0; burn_drive_free_subs(d); --subs_allocated; return 1; }``` |

Table 6: Pseudo code in stripped binary and corresponding refined pseudo code.

To intuitively demonstrate the effect of *refined pseudo code*, we provide a concrete example in Table 6. In real world strip scenario, the pseudo code decompiled from assembly code often lacks descriptive function and variable names and instead uses placeholders such as *"sub_E6D18", "dword_162354"*. To narrow the gap between pseudo code and natural language, we utilized the fine-tuned CodeT5 to recover meaningful names and strings, such as *"burn_drive_free_subs", "subs_allocated"*, which provide additional semantic information, even though the recovered strings may not be entirely accurate.

### 6 Conclusion

In this paper, we propose the CP-BCS framework, a novel approach that makes use of the control flow graph and pseudo code guidance. We manually construct the corresponding dataset that takes into account real-world scenarios. Finally, extensive experiments, ablation studies, and human evaluations demonstrate the effectiveness of CP-BCS. In practical applications, CP-BCS can significantly aid reverse engineers and security analysts in efficiently comprehending assembly code. We hope that our work can serve as a baseline while further prompting the development of this field.

### Limitations

Although our approach has been proven effective, it does not take into account code obfuscation (Menguy et al., 2021; Schloegel et al., 2022). Code obfuscation is a technique that alters the structure and logic of a program's code to make it difficult to analyze, preventing malicious actors from obtaining sensitive information or exploiting its vulnerabilities. We treat code obfuscation as an orthogonal problem, and any progress made in addressing it would be complementary to our approach.

### Acknowledgements

This work was partly supported by NSFC under No.62102360, CNKLSTISS, the Fundamental Research Funds for the Central Universities (Zhejiang University NGICS Platform), and the advanced computing resources provided by the Supercomputing Center of Hangzhou City University.

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

## A  LLMs on Assembly Code

Considering the emergence of large language models (LLMs), such as ChatGPT (OpenAI, 2022), we made an initial attempt to explore their potential in understanding assembly code. Through numerous attempts, we discover that LLMs generally possess only a rudimentary understanding of assembly code, such as memory operations and conditional jumps, as shown in Figure 6, without any higher-level abstract semantic comprehension.

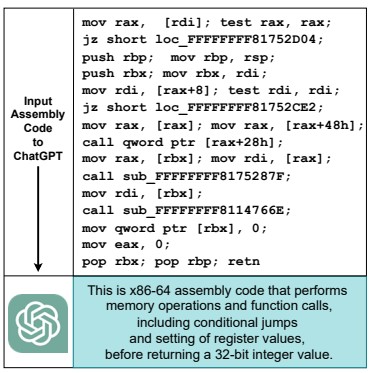

Figure 6: Inputting assembly code into ChatGPT.

## B  Preliminary Survey

We conduct a preliminary investigation that aims to explore the types of binaries that reverse engineers encounter in their daily work, the binaries that have impeded their process, and the specific components that they are most concerned with during the reverse engineering process. We conduct a survey with 15 reverse engineers from academia and industry and analyzed the collected data using descriptive statistics and content analysis. Our findings indicate that reverse engineers face a diverse range of binary programs, including both open-source and proprietary software, and encounter various challenges that affect their productivity and effectiveness. The most common types of binaries reported by participants were operating system utilities, drivers, and libraries. Regarding the specific components that reverse engineers are most concerned with during the reverse engineering process, our survey revealed that system-level functions, as well as networking and cryptography-related components, are the most frequently cited ones.

## C  Binary Projects

Table 7 displays a list of 51 binary projects and their corresponding versions.

| Binary Projects | Version | Binary Projects | Version |
|---|---|---|---|
| a2ps | 4.14 | binutils | 2.30 |
| bool | 0.2.2 | ccd2cue | 0.5 |
| cflow | 1.5 | coreutils | 8.29 |
| cpio | 2.12 | cppi | 1.18 |
| dap | 3.10 | datamash | 1.3 |
| direvent | 5.1 | enscript | 1.6.6 |
| findutils | 4.6.0 | gawk | 4.2.1 |
| gcal | 4.1 | gdbm | 1.15 |
| glpk | 4.65 | gmp | 6.1.2 |
| gnudos | 1.11.4 | grep | 3.1 |
| gsasl | 1.8.0 | gsl | 2.5 |
| gss | 1.0.3 | gzip | 1.9 |
| hello | 2.10 | inetutils | 1.9.4 |
| libiconv | 1.15 | libidn2 | 2.0.5 |
| libmicrohttpd | 0.9.59 | libosip2 | 5.0.0 |
| libtasn1 | 4.13 | libtool | 2.4.6 |
| libunistring | 0.9.10 | lightning | 2.1.2 |
| macchanger | 1.6.0 | nettle | 3.4 |
| patch | 2.7.6 | plotutils | 2.6 |
| readline | 7.0 | recutils | 1.7 |
| sed | 4.5 | sharutils | 4.15.2 |
| spell | 1.1 | tar | 1.30 |
| texinof | 6.5 | time | 1.9 |
| units | 2.16 | vmlinux | 4.1.52 |
| wdiff | 1.2.2 | which | 2.21 |
| xorriso | 1.4.8 | - | - |

Table 7: The 51 binary projects and versions.

## D Dataset Statistics and Explanations

Table 8 displays the statistics of our dataset on the X86 architecture for the three optimization levels. Table 9 displays the statistics of our dataset on the ARM architecture for the three optimization levels.

Currently, the dataset we've constructed is around the scale of 14k, and each sample has 9 different variants (across three computer architectures and three optimization options), leading to a total dataset size exceeding 100k. Compared to the source code summarization tasks where data collection is easier, the widely-used Java (Hu et al., 2018b) and Python (Wan et al., 2018) datasets have sizes of 70k and 80k, respectively. Although our dataset for a single architecture and single optimization option might appear smaller in comparison, there isn't a considerable difference in the order of magnitude. Notably, our collected binary projects are diverse, encompassing domains such as operating systems, databases, and networking. Additionally, it's important to highlight that the assembly of our dataset necessitates manual compilation—a process that is both rigorous and time-intensive.

| Datasets (Arch: X86) | O1 | O2 | O3 |
|---|---|---|---|
| Train | 12,937 | 12,338 | 11,249 |
| Validation | 1,617 | 1,542 | 1,406 |
| Test | 1,617 | 1,542 | 1,406 |
| Assembly Code: Avg. tokens | 234.00 | 244.10 | 346.74 |
| BI-CFG: Avg. nodes | 39.46 | 41.38 | 51.88 |
| BI-CFG: Avg. edges | 63.94 | 70.37 | 94.92 |
| Pseudo Code: Avg. tokens | 203.30 | 222.62 | 332.74 |
| Summary: Avg. tokens | 9.66 | 9.67 | 9.59 |

Table 8: Dataset statistics for X86 architecture with (O1, O2, and O3) optimization levels.

| Datasets (Arch: ARM) | O1 | O2 | O3 |
|---|---|---|---|
| Train | 7,453 | 6,839 | 5,963 |
| Validation | 932 | 855 | 745 |
| Test | 932 | 854 | 745 |
| Assembly Code: Avg. tokens | 276.87 | 279.26 | 390.37 |
| BI-CFG: Avg. nodes | 37.15 | 40.49 | 51.76 |
| BI-CFG: Avg. edges | 58.96 | 67.42 | 90.34 |
| Pseudo Code: Avg. tokens | 241.36 | 269.40 | 387.84 |
| Summary: Avg. tokens | 10.11 | 10.27 | 10.18 |

Table 9: Dataset statistics for ARM architecture with (O1, O2, and O3) optimization levels.

## E Human Evaluation

For our human evaluation, we invited 3 PhD students and 7 reverse engineers as volunteers. All of our volunteers have at least 1-3 years of experience in software engineering and reverse engineering. We randomly selected 200 examples from the dataset for volunteers to evaluate. The volunteers are required to answer the following questions.

- **Similarity**: How similar are the generated summary and ground-truth?
- **Fluency**: Is this generated summary syntactically correct and fluent?
- **Time-Cost**: The time and effort required to understand assembly functions.

For **Similarity** and **Fluency** metric, the rating scale is from 1 to 5, where a higher score means better quality. For **Time-Cost** metric, we divide assembly code samples into five groups, each corresponding to one of the following scenarios: "Assembly Code Only", "Pseudo Code (CodeT5+)", "None", "Function Name", and "CP-BCS", as shown in the Figure 5. There are no duplicates in the assembly code samples between any of the groups. We calculate the average time required by each volunteer to comprehend each group of assembly code samples. To ensure fairness, we attempt to maintain the same

| Arch: ARM | O1 | | | O2 | | | O3 | | |
|---|---|---|---|---|---|---|---|---|---|
| | **BLEU** | **ROUGE-L** | **METEOR** | **BLEU** | **ROUGE-L** | **METEOR** | **BLEU** | **ROUGE-L** | **METEOR** |
| Assembly | 26.24 | 23.09 | 13.84 | 27.55 | 24.44 | 14.85 | 24.14 | 21.18 | 12.14 |
| Pseudo (CodeT5) | 24.66 | 24.25 | 13.44 | 25.22 | 25.20 | 13.00 | 23.97 | 22.90 | 13.33 |
| CP-BCS w/o pseudo | 28.24 | 25.64 | 15.56 | 29.01 | 26.19 | 14.89 | 25.45 | 22.48 | 13.52 |
| CP-BCS w/o BI-CFG | 28.82 | 26.48 | 16.30 | 28.67 | 26.56 | 15.56 | 24.76 | 21.86 | 12.59 |
| CP-BCS w/o Refine | 29.13 | 27.10 | 15.58 | **29.84** | **27.77** | **16.30** | 25.59 | 23.38 | 13.48 |
| **CP-BCS** | **29.75** | **27.84** | **16.81** | 29.56 | 27.67 | 15.98 | **26.66** | **24.26** | **14.03** |

Table 10: Baselines and ablation study results on the dataset for ARM architecture.

| Arch: X86 | O1 | | | O2 | | | O3 | | |
|---|---|---|---|---|---|---|---|---|---|
| | **BLEU** | **ROUGE-L** | **METEOR** | **BLEU** | **ROUGE-L** | **METEOR** | **BLEU** | **ROUGE-L** | **METEOR** |
| Assembly | 21.69 | 16.98 | 9.64 | 18.24 | 12.72 | 6.64 | 21.52 | 17.48 | 9.27 |
| Pseudo (CodeT5) | 23.83 | 23.73 | 12.45 | 21.73 | 20.93 | 10.97 | 22.42 | 22.11 | 11.19 |
| CP-BCS w/o pseudo | 24.59 | 21.61 | 12.13 | 24.59 | 21.30 | 12.55 | 24.57 | 22.02 | 11.93 |
| CP-BCS w/o BI-CFG | 24.61 | 21.92 | 12.27 | 24.44 | 21.83 | 12.32 | 24.52 | 21.68 | 11.79 |
| CP-BCS w/o Refine | 25.66 | 23.41 | 12.96 | 25.53 | 23.64 | 13.31 | 25.56 | 23.79 | 12.49 |
| **CP-BCS** | **26.57** | **25.04** | **13.50** | **25.74** | **23.74** | **13.34** | **26.38** | **25.04** | **13.24** |

Table 11: Baselines and ablation study results on the dataset for X86 architecture.

| Arch: X64 | O1 | | | O2 | | | O3 | | |
|---|---|---|---|---|---|---|---|---|---|
| | **BLEU** | **ROUGE-L** | **METEOR** | **BLEU** | **ROUGE-L** | **METEOR** | **BLEU** | **ROUGE-L** | **METEOR** |
| Assembly | 22.88 | 18.82 | 11.09 | 21.52 | 16.96 | 8.77 | 22.53 | 19.03 | 11.01 |
| Pseudo (CodeT5) | 22.89 | 22.04 | 11.89 | 22.37 | 21.18 | 10.90 | 20.95 | 19.76 | 10.08 |
| CP-BCS w/o pseudo | 24.50 | 21.54 | 12.35 | 23.76 | 20.24 | 10.58 | 23.83 | 21.20 | 12.19 |
| CP-BCS w/o BI-CFG | 24.37 | 21.75 | 12.53 | 24.40 | 20.87 | 11.06 | 24.36 | 22.23 | 12.45 |
| CP-BCS w/o Refine | 25.61 | 23.20 | 13.12 | 24.96 | 21.98 | 11.89 | 24.31 | 21.78 | 12.52 |
| **CP-BCS** | **26.86** | **26.62** | **14.59** | **25.50** | **23.64** | **12.70** | **25.14** | **23.92** | **13.30** |

Table 12: Baselines and ablation study results on the dataset for X64 architecture.

number and length of assembly code instructions across all groups of samples as much as possible.

| ARCH:X86; OPT:O1 | BLEU | ROUGL-L | METEOR |
|---|---|---|---|
| CP-BCS | 26.57 | 25.04 | 13.50 |
| CP-BCS *on new test set* | 25.69 | 24.98 | 13.38 |

Table 13: Scalability Evaluations.

the Table 13, demonstrate that CP-BCS on new test set maintained similar performance, underscoring its scalability.

# F   Detailed Experimental Results

Table 10, Table 11 and Table 12 show the experiment results of CP-BCS for three different architectures and three different optimization levels.

To further demonstrate the scalability of CP-BCS, we conducted evaluations on approximately 200 newly compiled binary functions on X86 architecture and O1 optimization level (referred to as CP-BCS *on new test set*). The results, presented in