# OpenReview forum: "CP-BCS: Binary Code Summarization Guided by Control Flow Graph and Pseudo Code"
_EMNLP/2023/Conference — EMNLP 2023 Main_

### Official Review · Reviewer_bpsr · 2023-08-01

**Soundness:** 4

**Excitement:**

2: Mediocre: This paper makes marginal contributions (vs non-contemporaneous work), so I would rather not see it in the conference.

**Missing References:**

In the Related Works section, there is a recent work on understanding and capturing semantic meaning from code through a program-derived semantics graph. The authors should reference this work in this section:
- (NeurIPS-CAP 2020) Software Language Comprehension using a Program-Derived Semantics Graph, https://arxiv.org/abs/2004.00768


**Paper Topic And Main Contributions:**

This work focuses on generating complete summaries for binary functions, particularly stripped binary. To fully exploit the semantics of assembly code, authors present a control flow graph and pseudo code guided binary code summarization model, CP-BCS. CP-BCS uses a bidirectional instruction-level control flow graph and pseudo code with expert knowledge to learn binary function execution behavior and logic semantics. The authors evaluate CP-BCS on three different binary optimization levels (O1, O2, and O3) for three different computer architectures (X86, X64, 025 and ARM).

**Questions For The Authors:**

- While different components of the model are described in the Methodology section, there should also be some motivation as to why an autoencoder based architecture is used e.g., why not consider an encoder only architecture with an classier such as MLP component?
- The size of the datasets are in the thousands, which seem to be quite small. As such, I wonder what the scalability of the proposed approach is. Can the authors provide a section regarding the time complexity of their model?

**Reasons To Accept:**

- The Introduction and Related Works sections present some summary of limitations of existing works, as well as contributions of CP-BCS.
- The Methodology portions partitions the various components of the architecture well, with components of the: assembly instruction encoder, BI-CFG Encoder, Pseudo Code Encoder, and Summary Decoder.
- Dataset statistics are provided for broad range of datasets and computer architecture

**Reasons To Reject:**

- There is lack of motivation as to why the problem of investigation is important. For example, the authors may consider answering questions like what the benefits are with generating complete summaries for binary functions from an efficiency or accuracy point of view.  Alternatively, what are the different applications that can be benefited with solving this problem.
- In the Methodology section, it would be helpful if the authors could introduce a Problem Formulation section to formally describe the problem to be solved and any relevant assumptions and constraints.

**Reproducibility:**

4: Could mostly reproduce the results, but there may be some variation because of sample variance or minor variations in their interpretation of the protocol or method.

**Reviewer Confidence:**

4: Quite sure. I tried to check the important points carefully. It's unlikely, though conceivable, that I missed something that should affect my ratings.

---

> ### Author Rebuttal · Authors · 2023-08-28
>
> **Thank you for the detailed and constructive comments. In the following, we will clarify the concerns point by point.**
>
> *Q1: There is lack of motivation as to why the problem of investigation is important. ······ Alternatively, what are the different applications that can be benefited with solving this problem.*
>
> A1: Thanks for your suggestion. In today's reverse engineering domain, analyzing assembly code is a daunting task for security experts and reverse engineers, as they also grapple with decrypting unknown protocols, dissecting embedded firmware behaviors, and identifying hidden vulnerabilities in software. Assembly code, being a low-level representation of machine instructions, typically lacks the readability and structure of high-level programming languages. This means that security analysts must decipher the meanings of these instructions to understand the behavior of the program. Therefore, there is a pressing need for tools that can generate clear and concise summaries of binary programs. Our CP-BCS has the ability to automatically generate human-readable natural language summaries for binary functions. This feature significantly streamlines the comprehension process for both reverse engineers and security analysts alike. Human evaluations have shown that, when using CP-BCS-generated summaries, the productivity of security analysts can increase by a significant factor of 9.7, as detailed in Section 5.4. Such an advancement holds promise for boosting efficiency across various applications, including bug detection in binary programs, binary code optimization, and root-cause analysis.
>
> *Q2: In the Methodology section, it would be helpful if the authors could introduce a Problem Formulation section to formally describe the problem to be solved and any relevant assumptions and constraints.*
>
> A2: Thanks for pointing out this issue. We will incorporate the following Problem Formulation into the paper for greater clarity.
>
> **Problem Formulation:**
>
> Given a binary function with $l_c$ assembly code tokens $T_c = (c_1, c_2, ···,c_{l_c})$, and its BI-CFG with $l_n$ nodes $T_n = (n_1, n_2, ···,n_{l_n})$ , and its corresponding pseudo code with $l_p$ tokens $T_p = (p_1, p_2, ···,p_{l_p})$, CP-BCS predicts the next summary tokens based on the existing tokens $T_s = (</s>, s_1, s_2, ···, s_{m-1},···)$，where $</s>$ is a special starting tag for summary input.
>
> The objective of our research is to generate a concise natural language summary for the binary function. Like the binary function shown in Figure 1 (*"mov rax, [rdi] test rax, ······, retn"*)，our target is to generate the natural language summary: *"free all resources associated with context_handle."*
>
> *Q3: While different components of the model are described in the Methodology section, there should also be some motivation as to why an autoencoder based architecture is used e.g., why not consider an encoder only architecture with an classier such as MLP component?*
>
> A3: Thank you for the comment about the model architecture. As discussed in the introduction section (line 66), considering that the binary code summarization task involves generating a natural language text summary, which is a generative task essentially, an encoder-only architecture is insufficient to achieve the task’s objectives. Thus, we have chosen an encoder-decoder architecture. The encoder component aims to comprehensively extract behavior and logic semantics across three different dimensions (assembly instruction, BI-CFG, and pseudo code). The decoder component is designed to generate corresponding summary tokens in an autoregressive manner.
>
> *Q4: The size of the datasets are in the thousands, which seem to be quite small. As such, I wonder what the scalability of the proposed approach is. Can the authors provide a section regarding the time complexity of their model?*
>
> A4: Thanks for highlighting these points. At present, the dataset we've constructed is around the scale of 14k, and each sample has 9 different variants (across three computer architectures and three optimization options), leading to a total dataset size exceeding 100k. Compared to the source code summarization tasks where data collection is easier, the widely-used Java[1] and Python[2] datasets have sizes of 70k and 80k, respectively. Although our dataset for a single architecture and single optimization option might appear smaller in comparison, there isn't a considerable difference in the order of magnitude. Notably, our collected binary projects are diverse, encompassing domains such as operating systems, databases, and networking. Additionally, it's important to highlight that the assembly of our dataset necessitates manual compilation—a process that is both rigorous and time-intensive. Concurrently, we're continuously building a larger dataset and will share it on the corresponding GitHub  repository soon.
>
> [1] Deep code comment generation, ICPC 2018.
>
> [2] Improving automatic source code summarization via deep reinforcement learning, ASE 2018.
>
> **Scalability:**
>
> To further demonstrate the scalability of our proposed approach, we conducted evaluations on approximately 200 newly compiled binary functions on X86 architecture and O1 optimization level (referred to as *CP-BCS on new test set*). The results, presented in the table below, demonstrate that CP-BCS on new test set maintained similar performance, highlighting its scalability.
>
> | Arch: X86, OPT: O1 | BLEU | ROUGE-L | METEOR |
> | --- | --- | --- | --- |
> | CP-BCS | 26.57 | 25.04 | 13.50 |
> | *CP-BCS on new test set* | 25.69 | 24.98 | 13.38 |
> ||
>
> **Time complexity:**
>
> The time complexity of CP-BCS model depends on three encoders (assembly instruction encoder, BI-CFG encoder, and pseudo code encoder) and one summary decoder.
>
> + For the assembly instruction encoder and pseudo code encoder, both employ the transformer encoder design. Assuming the length of the assembly code length and pseudo code length are both $n$ and the feature dimension is $d$,  their time complexities stand at $O(n^2d)$.
> + The BI-CFG encoder, based on a graph attention network, has its complexity determined by the number of nodes  $v$ and edges $e$ in each binary function, with a node feature dimension of  $d$. Its complexity is  $O(vd^2+ed)$.
> + The summary decoder’s predominant time complexity arises from its three cross-attention modules, making its time complexity $O(2*n^2d+v^2d)$.
>
> In our empirical evaluations, generating summaries for 1,000 test samples took around 33 seconds, which we consider to be highly efficient and capable of meeting practical requirements.
>
> *Q5: Missing References*
>
> A5: Thanks for pointing out this issue. We promise to include this reference “Software Language Comprehension using a Program-Derived Semantics Graph (NeurIPS-CAP 2020)” in the Related Works section.

---

### Official Review · Reviewer_9guV · 2023-08-04

**Soundness:** 3

**Excitement:**

3: Ambivalent: It has merits (e.g., it reports state-of-the-art results, the idea is nice), but there are key weaknesses (e.g., it describes incremental work), and it can significantly benefit from another round of revision. However, I won't object to accepting it if my co-reviewers champion it.

**Paper Topic And Main Contributions:**


This paper proposes a control flow graph and pseudo code guided binary code summarization method, which extracts the execution behavior and semantics of assembly code. The proposed model utilizes three different encoders to model assembly instruction, BI-CFG and pseudo code, separately, and employs a decoder to generate the final summarization. They also construct and release a binary code summarization data including assembly code and summary pairs. Experiments with automatic metrics and human evaluation demonstrate the effectiveness of the proposed method.


**Reasons To Accept:**


1. This paper constructs and releases a comprehensive assembly code summarization dataset. It will facilitate the development of this direction.

2. They design a multi-source encoder framework to make full use of the control flow graph and pseudo code guidance in binary code summarization.


**Reasons To Reject:**


1. The dataset constructed by this paper is relatively small. It would be better to build a large dataset by some semi-supervised methods.

2. Other SOTA code summarization methods, like using pre-trained models, should be compared in the experiments.


**Reproducibility:**

3: Could reproduce the results with some difficulty. The settings of parameters are underspecified or subjectively determined; the training/evaluation data are not widely available.

**Reviewer Confidence:**

4: Quite sure. I tried to check the important points carefully. It's unlikely, though conceivable, that I missed something that should affect my ratings.

---

> ### Author Rebuttal · Authors · 2023-08-28
>
> **Thank you for the detailed and constructive comments. In the following, we will clarify the concerns point by point.**
>
> *Q1: The dataset constructed by this paper is relatively small. It would be better to build a large dataset by some semi-supervised methods.*
>
> A1: Thanks for pointing this out. Currently, the dataset we've constructed is around the scale of 14k, and each sample has 9 different variants (across three computer architectures and three optimization options), leading to a total dataset size exceeding 100k. Compared to the source code summarization tasks where data collection is easier, the widely-used Java[1] and Python[2] datasets have sizes of 70k and 80k, respectively. Although our dataset for a single architecture and single optimization option might appear smaller in comparison, there isn't a considerable difference in the order of magnitude. Moreover, to ensure our dataset is compatible across the three computer architectures and three compilation optimization options, we have to perform manual compilation and choose the appropriate compilation configurations. At this current stage, this remains essential.  At the same time, we're continuously collecting and building a larger dataset. In line with your recommendation, we're delving into semi-supervised strategies, such as presetting configuration parameters. Once completed, we'll make it open-source on the corresponding GitHub repository to facilitate further research.
>
> [1] Deep code comment generation, ICPC 2018.
>
> [2] Improving automatic source code summarization via deep reinforcement learning, ASE 2018.
>
> *Q2: Other SOTA code summarization methods, like using pre-trained models, should be compared in the experiments.*
>
> A2：Thanks for the comment. We employed code-related pre-trained models (CodeT5 [1], CodeT5+ [2], and UniXcoder [3]) for fine-tuning to compare with CP-BCS. The experimental results (on X64 architecture and O1 optimization level), as shown in the table below, indicate that pre-trained models perform well, with a notably better performance on pseudo code than assembly code. Intuitively, this makes sense as pseudo code is more akin to high-level programming languages compared to assembly code. Overall, our proposed CP-BCS demonstrates a superior performance compared to direct fine-tuning of pre-trained models.
>
> | Arch: X64, OPT: O1 | BLEU | ROUGE_L | METEOR |
> | --- | --- | --- | --- |
> | assembly code (CodeT5-base) | 21.87 | 20.78 | 11.05 |
> | pseudo code (CodeT5-base) | 22.89 | 22.04 | 11.89 |
> | assembly code (CodeT5+, 220M) | 23.07 | 21.02 | 11.58 |
> | pseudo code (CodeT5+, 220M) | 24.14 | 23.83 | 12.48 |
> | assembly code (Unixcoder) | 22.01 | 18.48 | 10.48 |
> | pseudo code (Unixcoder) | 23.17 | 22.65 | 12.35 |
> | CP-BCS | **26.86** | **26.62** | **14.59** |
> ||
>
> We will include the experimental outcomes in the Experiments Section of the paper to enhance the comprehensiveness and soundness of our results.
>
> [1] CodeT5: Identifier-aware Unified Pre-trained Encoder-Decoder Models for Code Understanding and Generation, EMNLP 2021
>
> [2] CodeT5+: Open Code Large Language Models for Code Understanding and Generation, arXiv 2023
>
> [3] UniXcoder: Unified Cross-Modal Pre-training for Code Representation, ACL 2022

---

### Official Review · Reviewer_hYNs · 2023-08-04

**Soundness:** 3

**Excitement:**

4: Strong: This paper deepens the understanding of some phenomenon or lowers the barriers to an existing research direction.

**Paper Topic And Main Contributions:**

This paper focuses on assembly code summarization generation task. The authors first construct an assembly code - summary paired dataset containing three computer systems with three optimization levels. Then they present a control flow graph and pseudo code guided  binary code summarization framework called CP-BCS. The empirical results show that CP-BCS has good performance and can greatly improve the efficiency of reverse engineering.

**Questions For The Authors:**

1. The title of the paper is binary code summarazation, but it is actually assembly code summarazation. Is that appropriate?
2. In a network structure, why is it pseudo code cross-attention, then assembly code cross-attention, and finally BI-CFG cross-attention?
3. What is the result of concatenating assembly code and pseudo code directly?

**Reasons To Accept:**

1. The first one proposes the binary code summarization task.
2. The authors construct a parallel dataset containing different system architectures.
3. CP-BCS method is obviously superior to baseline models and can improve the efficiency of reverse engineering.


**Reasons To Reject:**

N/A

**Reproducibility:**

3: Could reproduce the results with some difficulty. The settings of parameters are underspecified or subjectively determined; the training/evaluation data are not widely available.

**Reviewer Confidence:**

4: Quite sure. I tried to check the important points carefully. It's unlikely, though conceivable, that I missed something that should affect my ratings.

---

> ### Author Rebuttal · Authors · 2023-08-28
>
> **Thank you for the detailed and constructive comments. In the following, we will clarify the concerns point by point.**
>
>
> *Q1: The title of the paper is binary code summarization, but it is actually assembly code summarization. Is that appropriate?*
>
> A1: Thanks for pointing it out. There are primarily two reasons for this matter. Firstly, our research focuses on binary closed-source programs. The initial step in analyzing such programs usually involves translating them into assembly code. Secondly, within the context of practical reverse engineering, both the academic and industrial communities frequently employ the term "binary code" to denote related research areas, like the domain of binary code similarity[1][2]. In actual binary code similarity tasks, the operation target is also assembly code. Due to these factors, we chose to utilize the phrasing "binary code summarization" in the title of our work.
>
> [1] Jtrans: Jump-aware transformer for binary code similarity detection, ISSTA 2022
>
> [2] Order matters: Semantic-aware neural networks for binary code similarity detection, AAAI 2020
>
> *Q2: In a network structure, why is it pseudo code cross-attention, then assembly code cross-attention, and finally BI-CFG cross-attention?*
>
> A2: Thanks for the comment. Our choice of this particular order is driven by its slightly superior performance. However, in practical scenarios,  different cross-attention orders have minimal impact on the final performance. We take the dataset of the X64 architecture and O1 optimization level for example to provide concrete results:
> | Three Cross-attention Orders | BLEU | ROUGE_L | METEOR |
> | --- | --- | --- | --- |
> | assembly code→BI-CFG→pseudo code | 26.71 | 26.37 | 14.45 |
> | assembly code→pseudo code→BI-CFG | 26.50 | 26.40 | 14.39 |
> | BI-CFG→assembly code→pseudo code | 26.45 | 25.95 | 14.31 |
> | BI-CFG→pseudo code→assembly code | 26.47 | 26.08 | 14.49 |
> | **pseudo code→assembly code→BI-CFG** *(in our paper)* | **26.86** | **26.62** | 14.59 |
> | pseudo code→BI-CFG→assembly code | 26.86 | 26.45 | **14.67** |
> ||
>
>
> Based on the experimental results, different orders only have a slight impact on the final performance (the BLEU score did not fluctuate by more than 0.5 points). We will incorporate the experimental results of different cross-attention orders into the Experiments Section of the paper to enhance the comprehensiveness and soundness of our results.
>
> *Q3. What is the result of concatenating assembly code and pseudo code directly?*
>
> A3: Thanks for the comment. Directly concatenating assembly code and pseudo code requires extending the maximum length of the encoder. We conducted experiments, and the results are as follows:
> | Arch: X64, OPT: O1 | BLEU | ROUGE_L | METEOR |
> | --- | --- | --- | --- |
> | concat(assem+pseudo) | 24.49 | 21.71 | 12.68 |
> | concat(assem+pseudo)+BI-CFG | 25.83 | 24.33 | 13.55 |
> | CP-BCS | **26.86** | **26.62** | **14.59** |
> ||
>
> Based on the experimental results, direct concatenation can degrade the model's final performance to some extent, as shown in the last two rows of the table. Therefore, assigning a separate encoder for assembly code and pseudo code is a better choice.

---

### Meta-Review · Area_Chair_NaAx · 2023-09-18

**Recommendation:** 5

**Metareview:**

This paper introduces a novel framework, CP-BCS (Control Flow Graph and Pseudo Code Guided Binary Code Summarization), for the task of assembly code summarization. The authors build a dataset containing assembly code-summary pairs across three computer systems with different optimization levels. CP-BCS uses a bidirectional instruction-level control flow graph and pseudo code to learn the execution behavior and logic semantics of binary functions. The method is evaluated on three different binary optimization levels for three different computer architectures, showing considerable improvement in the efficiency of reverse engineering.

Main Contributions:

The paper introduces the task of binary code summarization.
The authors construct a parallel dataset with assembly code-summary pairs across different system architectures.
A new framework, CP-BCS, is proposed, which uses a multi-source encoder strategy to leverage control flow graphs and pseudo code for binary code summarization.
The authors provide comprehensive dataset statistics across a broad range of datasets and computer architectures.

Reasons for Acceptance:

The introduction of the binary code summarization task fills a gap in current research.
The construction and release of a comprehensive assembly code summarization dataset will facilitate further development in this field.
The CP-BCS method shows superior performance to baseline models, improving the efficiency of reverse engineering.
The methodology is well-described and partitioned into various components, including the assembly instruction encoder, BI-CFG Encoder, Pseudo Code Encoder, and Summary Decoder.
The paper provides a thorough evaluation of CP-BCS on different binary optimization levels and computer architectures.

---

### Decision · Program_Chairs · 2023-10-07

**Decision:**

Accept-Main

**Comment:**

This paper introduces a novel framework, CP-BCS (Control Flow Graph and Pseudo Code Guided Binary Code Summarization), for the task of assembly code summarization. The authors build a dataset containing assembly code-summary pairs across three computer systems with different optimization levels. CP-BCS uses a bidirectional instruction-level control flow graph and pseudo code to learn the execution behavior and logic semantics of binary functions. The method is evaluated on three different binary optimization levels for three different computer architectures, showing considerable improvement in the efficiency of reverse engineering.

Main Contributions:

The paper introduces the task of binary code summarization.
The authors construct a parallel dataset with assembly code-summary pairs across different system architectures.
A new framework, CP-BCS, is proposed, which uses a multi-source encoder strategy to leverage control flow graphs and pseudo code for binary code summarization.
The authors provide comprehensive dataset statistics across a broad range of datasets and computer architectures.

Reasons for Acceptance:

The introduction of the binary code summarization task fills a gap in current research.
The construction and release of a comprehensive assembly code summarization dataset will facilitate further development in this field.
The CP-BCS method shows superior performance to baseline models, improving the efficiency of reverse engineering.
The methodology is well-described and partitioned into various components, including the assembly instruction encoder, BI-CFG Encoder, Pseudo Code Encoder, and Summary Decoder.
The paper provides a thorough evaluation of CP-BCS on different binary optimization levels and computer architectures.